# Bidirectional, Analog Current Source Benchmarked with Gray Molasses-Assisted Stray Magnetic Field Compensation

**Jakub Dobosz** *[iD], **Mateusz Bocheński** and **Mariusz Semczuk** [iD]

Institute of Experimental Physics, University of Warsaw, Pasteura 5, 02-093 Warsaw, Poland;
mbochenski104@gmail.com (M.B.); msemczuk@fuw.edu.pl (M.S.)
*   Correspondence: j.dobosz2@uw.edu.pl

**Abstract:** In ultracold-atom and ion experiments, flexible control of the direction and amplitude of a uniform magnetic field is necessary. It is achieved almost exclusively by controlling the current flowing through coils surrounding the experimental chamber. Here, we present the design and characterization of a modular, analog electronic circuit that enables three-dimensional control of a magnetic field via the amplitude and direction of a current flowing through three perpendicular pairs of coils. Each pair is controlled by one module, and we are able to continuously change the current flowing thorough the coils in the $\pm 4$ A range using analog waveforms such that smooth crossing through zero as the current's direction changes is possible. With the electrical current stability at the $10^{-5}$ level, the designed circuit enables state-of-the-art ultracold experiments. As a benchmark, we use the circuit to compensate stray magnetic fields that hinder efficient sub-Doppler cooling of alkali atoms in gray molasses. We demonstrate how such compensation can be achieved without actually measuring the stray fields present, thus speeding up the process of optimization of various laser cooling stages.

**Keywords:** stray magnetic fields; compensation coils; sub-Doppler cooling; bipolar current source; bidirectional current source

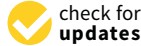



## 1. Introduction

The ability to create a stable magnetic field at the location of an atomic sample is the backbone of many modern ultracold experiments [1–4]. This is almost exclusively achieved by running current through properly arranged coils that are used to create two types of magnetic field distributions of particular importance: a quadrupole and uniform magnetic field distributions. For this purpose, many low-noise current source designs have been developed. In many applications, such as forming a magnetic trap [5] or controlling interactions with Feshbach resonances [6], a unipolar current source is sufficient [7–10]. However, for the full control of cold samples, the ability to create a uniform magnetic field of user-defined amplitude and direction is required. In principle, it can be achieved with unipolar current sources in conjunction with an H-bridge, but this solution does not provide continuous crossing through zero current. Such a feature is useful, for example, when, during an experimental sequence, the magnetic field direction needs to be adiabatically changed to orient the quantization axis differently than during the optical pumping stage. Bidirectional current sources, on the other hand, enable both the direction and the amplitude control of the current and, if properly designed, offer a smooth zero-current crossing [11–15]. With such sources, running current through each pair of three mutually orthogonal pairs of coils in a near-Helmholtz configuration provides the ability to create uniform magnetic field waveforms. As a result, during an experimental sequence, the magnetic field vector can, for example, rotate or oscillate around 0 G.

We present a compact, fully analog design of a bidirectional current source that can control currents up to 4 A (see Supplementary Materials for source files and detailed schematics). It is stabilized with proportional-integral-derivative (PID) controllers and

has a low noise at the $10^{-5}$ level, sufficient for many modern experiments. The circuit operates with single-sided supplies, which makes the overall setup more compact and cost effective than designs powered with symmetric supplies. Of the two frequently used approaches, digital and analog, we choose the latter one. Digital circuits, although offering better independence of nonlinearities, thermal drifts, and repeatability [16], often rely on expensive, complex, high-resolution digital-to analog (DAC) and analog-to-digital (ADC) converters [12,13]. Moreover, switching of digital signals is a well known noise source for the analog part of the circuit. Analog devices, on the other hand, need careful compensation of quiescent currents and parasitic impedances, but they allow construction of more compact devices without the need for separation of analog and digital parts. They are also easier to make and more intuitive to debug by an inexperienced user, a feature quite useful in particular in an academic environment, where new students join research groups often without too much hands-on experience in electronics. The analog circuit presented here allows for the control of current within its full range with a bandwidth up to 1 kHz and can be used for most applications common in ultracold atomic physics, such as canceling stray magnetic fields, defining a quantization axis, or splitting Zeeman sub-levels for state transfer with Raman beams or microwaves.

As an element of validation of the quality of the designed current source, we propose and demonstrate a fast and robust method of compensating stray magnetic fields using sub-Doppler cooling in gray molasses, which has been shown to work the best when the magnetic field at the location of atoms is near 0 G [17–21]. We postulate that the circuit is sufficiently good for many demanding cold-atom experiments if it enables repeatable and efficient sub-Doppler cooling, defined by the lowest temperatures that can be reached in a given experiment. For the demonstration of our stray field compensation method, we use D1-line gray molasses cooling of $^{39}$K, but the technique can be applied to any species where cooling relies on coherences between hyperfine levels of the ground state.

The key advantage of our approach is its speed. The actual value of the stray magnetic fields present in the experimental chamber is impossible to measure without putting a sensor at the location of atoms, but any method requires an initial estimation of this value to obtain a signal to optimize. Our method relaxes this requirement to the extent that the initial guess sufficient to start the optimization can be obtained even by measuring the field close to the vacuum chamber with a smartphone. In our specific case, the initial guess that was further refined during the cancellation procedure was 420 mG, whereas the actual value of stray fields in the experimental setup approaches 700 mG. Overall, within 2 h of data collection, we minimized stray fields to enable cooling of $^{39}$K atoms to 8 μK, which is in line with the best known results obtained with carefully compensated stray fields by Salomon et al. [19] (6 μK) and by Nath et al. [22] (12 μK).

## 2. Circuit Design

A module of the bidirectional current source consists of several parts forming a loop (Figure 1). The source part pushes the current through external coils and a high-power current sense resistor $R_{\text{sense}}$. A process variable—current—is measured on the sense resistor in the current measurement section and is sent to the last stage, the PID controller. In the PID part, the value of the measured current is compared with an analog set point, and a control signal is calculated. It is then sent to the source part closing the feedback loop.

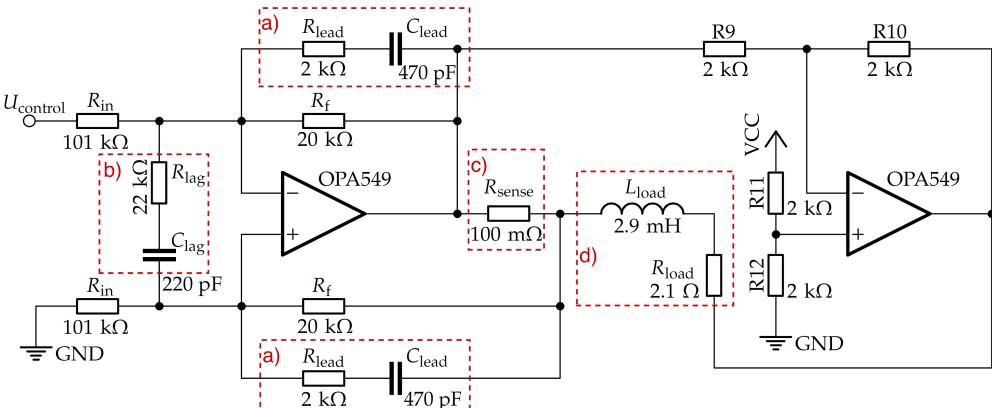

**Figure 1.** The source section of the push–pull operational amplifier bipolar current source. The main operational amplifiers (OPA549) work interchangeably as a current source and a sink allowing for bidirectional current flow through external coils. The current value is sensed on the $R_{sense}$ resistor, placed in series with $L_{load}$. The output current is proportional to the input $U_{control}$ voltage referenced to the second input (here, grounded). For the full formula, see Equation (1). (**a**) Parts of the lead compensator for response timing enhancement. (**b**) Parts of the optional lag compensator for enhancing the steady state performance. (**c**) The high power, temperature-stable sense resistor for current measurement. (**d**) External coils with total inductance $L_{load} = 2.9\,\text{mH}$ and total resistance $R_{load} = 2.1\,\Omega$.

There are several common circuits for bipolar current sources, such as Howland pumps, current-in/current-out amplifiers, and push–pull setups (based on transistors or operational amplifiers) [11–15]. We choose the latter solution to obtain high-output currents and smooth, linear zero-current crossing. The other listed designs cannot easily fulfill both requirements. The current source part is a modified version of a design originally developed by Nicolas Arango and Irene Kuang [23].

The core of the circuit is a pair of high-power amplifiers (OPA549, Texas Instruments), which are capable of sourcing up to 8 A continuously. They are connected in a push–pull configuration with the 2 A/V gain. The current flows between the outputs of the amplifiers, through a sense resistor and external coils.

To enhance the timing characteristic of the current source, especially when it is loaded with an inductive coil, we added lead compensation. It may be adjusted for a particular load impedance by changing the capacitor and resistor values. The lead compensation lowers both the rise time and the maximum overshoot by adding an additional pole in the transfer function on the high frequency side. The elements of the lead compensator need to be placed symmetrically in feedback loops for both amplifiers inputs.

An optional lag compensation enhances steady-state stability. To a large extent, it is already provided by the PID control section, but it is useful in noisy environments, since it directly low passes the inputs of the current source section.

The lead and lag compensators set the primary bandwidth limitations of the whole circuit, since the PID parameters are adjusted to the performance of the current source section. However, the transfer function of the system may be modified to enhance the circuit's bandwidth by changing the values of the resistors and capacitors used in both compensators.

The output current is proportional to the input voltage, $U_{control}$, referenced to the second input, which is shorted to the ground. This makes the circuit's response symmetric around 0 V, and, as such, it provides an intuitive relation between the direction and the amplitude of the current. The control signal may be also referenced if controlled with a digital device with a different operation voltage.

Within the range of unity gain of the transfer function, the current flowing through the coils can be expressed as

$$I = U_{\text{control}} \frac{R_{\text{f}}}{R_{\text{in}} R_{\text{sense}}} \tag{1}$$

where resistances $R_{\text{f}}$, $R_{\text{in}}$ and $R_{\text{sense}}$ 470 pF refer to the feedback resistor, the input resistor, and the sense resistor, respectively (see Figure 1). A bidirectional current shunt monitor (INA170, Texas Instruments) is used to measure the current flowing through the sense resistor. We choose this particular part for its high common mode rejection, high voltage input, and low output noise density of $20 \text{ pA}/\sqrt{\text{Hz}}$. This serves both as a process variable input for the PID feedback loop and as an output for current monitoring. The current monitor output is chosen to be within the 0–2.5 V range (symmetric around 1.25 V reference voltage) such that it can be easily measured with a microcontroller referenced to 2.5 V.

To reach the stability of the current of the order of tens of ppm, we control the current source section with a PID controller. In the controller's design, we use OP07C and OPA2227 amplifiers by Texas Instruments with 0.6 MHz and 8 MHz bandwidths, respectively. The measured PID transfer function rolls off at 1 kHz. Although the OPA549 amplifiers can operate at higher frequencies, the stability requirements and high output current range put constraints on the transfer function of the system. For this reason, the total roll off frequency is lower than for individual elements.

The user-defined set point is fed into the circuit via a differential amplifier (AD8421, Analog Devices) to ensure isolation between the board and the environment. The resistors and capacitors in the PID feedback loop are placed in DIP sockets to allow easy modification of the parameters of the PID. For the integral part, mainly responsible for the response time, the capacitors can be changed with a piano DIP-switch located in the easily accessible front part of the board. All PID parts are built around one operational amplifier. This solution intertwines the dependence of the PID settings on electronic components, but it makes the circuit more compact and less prone to interference with other parts of the circuit due to a minimized number of electronic elements and traces. Due to the linearity of functional sections of the PID, the one-amplifier solution operates in the same manner as separate amplifiers for individual PID sections. An additional amplifier is added to provide variable gain of the total control signal, which is fed into the current source section. We account for potential dark or parasitic currents by including a potentiometer to set an offset level to the control signal.

The device is powered from external, linear power supplies available in most laboratories. The push–pull configuration uses operational amplifiers that can be single supplied. The direction of the current flowing through the system depends on the relative levels of voltages at the outputs of the amplifiers; thus, there is no change in the voltage polarization when the current polarization changes. The amplifiers only switch between the sink and source modes. The PID, current sense, and logical parts require a 24 V power supply. The current source part uses a separate 18 V supply to minimize unnecessary heating due to voltage drops on the internal output transistors. When used at lower currents or with external cooling, the setup may be powered from one source.

## 3. Circuit Performance

To measure the noise level we use a 12-bit oscilloscope (Teledyne Lecroy, model HDO 4054) to acquire a 10 s long, continuous traces from the monitor output. We also verify the consistency of the noise characterization using a high-accuracy current transducer (IT 400-S Ultrastab, LEM). Both methods appear to be comparable except for currents of the order of 100 mA and less so where the signal-to-noise ratio of the transducer measurements is much worse than that for the monitor output. This is caused by the low sensitivity at low currents of the available transducer, which is optimized for current readouts in the range of several amperes.

We characterize the circuit using linear spectral density (LSD), which is derived from the Fourier transform (FFT) of a long-run signal measured with an oscilloscope and the

relations $\text{LSD}^2 = \text{PSD} \sim |\text{FFT}|^2$, where PSD stands for power spectral density. The measurement is taken with a 2.5 MHz sampling rate, so the cutoff of the Fourier transform, due to the Nyquist theorem, equals 1.25 MHz. The noise floor trace is acquired with the current source (both power supplies) turned off.

The LSD noise is measured for several currents and is shown in Figure 2a for the maximum design current of 4 A and for the current corresponding to the value used for stray field compensation along the y-direction. For all investigated currents, the measured noise level is of the order of tens of $\mu\text{A}/\sqrt{\text{Hz}}$ with 45 ppm of integrated noise at 4 A (the noise integrated over a 1 MHz bandwidth).

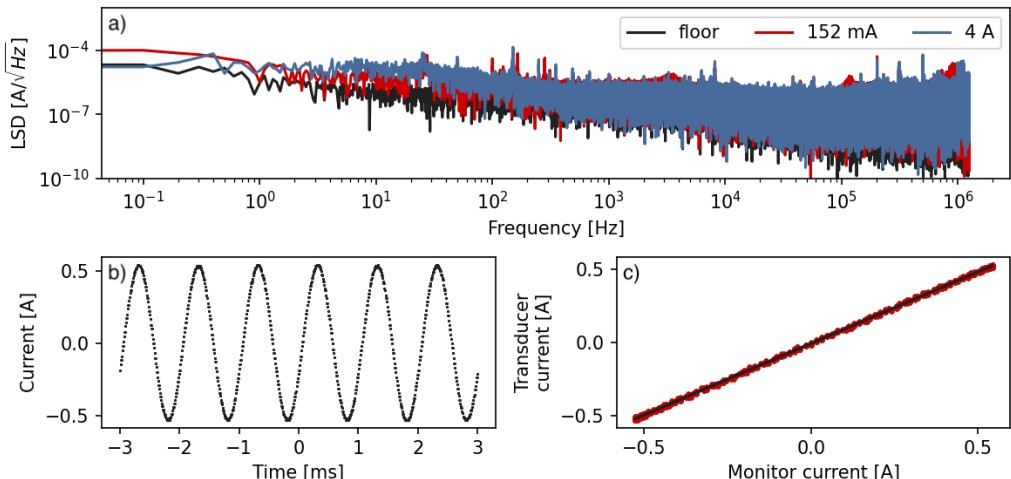

**Figure 2.** (**a**) The linear spectral density for chosen output currents measured at 100 Hz RBW in the range up to 1 MHz. The red line shows the noise level for the *y*-axis coils at a current used for compensating the stray field. The blue line corresponds to the maximum current supported by the designed source. The stability measurement is averaged over a period of 1 min; (**b**) The current in the coils is measured indirectly on the monitor output of the current supply. The circuit is driven by a 1 kHz sinusoidal input signal ranging from $-0.5$ A to 0.5 A. The zero crossing is smooth due to push–pull operational amplifier configuration; (**c**) A parametric plot of the current measured directly with a high-accuracy current transducer vs. the current measured indirectly on the monitor output of the current supply while the circuit is driven as in (**b**). The line is a linear fit with a slope of 1.0179(4).

The bandwidth and the step response depend both on the PID settings and the current value. For the best performance, the resistors and capacitors in the PID loop need to be matched to the inductive and resistive load of the coils. The maximum continuous current of the power amplifiers can reach $\pm 8$ A, but we deliberately limit it to $\pm 4$ A, which is sufficient for producing magnetic fields needed for most cold-atom applications with the set of coils we currently use. Even though the setup is fully analog and, in principle, can source any arbitrary current within its operational range, it is controlled with a DAC with limited resolution. Thus, the $\pm 4$ A limit increases the resolution of the current settings.

The DAC we use is an NI PXI-6733 card, which is an integral part of our experiment control system. It has a 16-bit resolution, which corresponds to a control voltage step of 0.3 mV. This translates to 0.54 mA current step or $\sim$2.4 mG step for the y-coil. This conversion takes into account the PID and the current source voltage-to-current coefficient of 1.81 A/V, which his determined experimentally by applying a known control voltage to the circuit and measuring the output current with a high-accuracy current transducer.

For the maximum current of 4 A, the roll-off frequency of the PID is 1 kHz. It is lower than both the simulated value ($\sim$5 kHz) and the value measured on a purely resistive load. This is, however, unsurprising, because the inductive load of the connected coils opposes current changes at both higher frequencies and higher current amplitudes, as it adds another pole to the PID transfer function. As an example of the ability to generate

waveforms with the circuit connected to one pair of coils, we produce a sinusoidal current modulation $I = 0.5 I_{\text{p-p}} \sin(2\pi f t)$ with $I_{\text{p-p}} = 1$ A and $f = 1$ kHz. The current is measured using the output monitor (Figure 2b) and a current transducer (IT 400-S Ultrastab, LEM). A parametric plot of the one measurement method vs. the other is shown in Figure 2c. The monitor output reliably represents the actual current supplied by the circuit, with a Pearson correlation coefficient between the measurement methods of 0.9998(6) and a slope of a fitted linear relation equal to 1.0179(4). We verify that a sinusoidal current modulation from $-0.5$ A to 0.5 A and 0 A to 1 A can be generated at frequencies up to 2 kHz before it becomes distorted.

The OPA549 power amplifiers have internal protection against overheating—once 125 °C is reached, they shut down and short E/S pins to the ground until they have cooled down. The thermal resistance of a bare amplifier equals $\theta_{\text{JA}} = 30$ °C/W, which makes them heat up quickly at high continuous currents. We assure proper operation of the devices for the entire design current range by adding a radiator with thermal resistance $\theta_{\text{HA}} = 6$ °C/W, which, in total, gives $\theta_{\text{JA}} = 7.9$ °C/W. We further increase the heat dissipation efficiency by adding a fan such that the temperature of the amplifier is always well below 125 °C. The power amplifiers are mounted at the edge of the circuit board such that a better (in terms of lower thermal resistance) or even water cooled radiator could be installed if currents >4 A are to be used. All noise measurements are performed with the cooling fan turned on.

## 4. Stray Fields Compensation

In cold-atom experiments, bipolar current sources are frequently used to cancel stray magnetic fields inside the experimental chamber and to define a quantization axis. Here, we demonstrate the functionality of the designed power supply by minimizing the total magnetic field present at the location of trapped atoms without actually measuring the net field. The quality of the stray fields cancellation is assessed based on the measurement of $^{39}$K temperature reached after a period of gray molasses cooling on the D1-line as a function of the applied magnetic field vector $\vec{B}$.

Among alkali atoms, $^{6,7}$Li and $^{39,41}$K have such a small $P_{3/2}$ state hyperfine splitting that sub-Doppler cooling on the D2-line is limited. Using the D1-line instead, which has a better resolved $P_{1/2}$ state hyperfine levels, has proven to be an efficient method of reaching lower temperatures. This comes at a price of using an additional laser setup operating on the D1 line, but the gain in phase space density over a standard D2-line magneto-optical trap is crucial for many experiments requiring large samples of degenerate quantum gases [24–26].

The combination of Sisyphus cooling and velocity-selective coherent population trapping present in D1-line gray molasses makes it possible to cool atoms well below 1/10 of the Doppler limit. The molasses are composed of two frequencies, each blue-detuned from either the $^{39}$K D1 cooling transition $|F = 2\rangle \rightarrow |F' = 2\rangle$ or the repumping transition $|F = 1\rangle \rightarrow |F' = 2\rangle$. When the detunings of the two lasers are such that their energy difference equals the energy difference between the ground state hyperfine levels $|F = 1\rangle$ and $|F = 2\rangle$, the Raman condition is fulfilled, and a set of long-lived dressed dark states (superposition of only the ground state hyperfine levels) is generated. They are accompanied by bright states that contain the $P_{1/2}$ component. Both the dark and the bright states vary throughout real space due to the presence of polarization gradients created by laser beams. The atoms that are initially in bright states lose their kinetic energy by climbing the potential hill before being pumped back into dark states, whereas the atoms in dark states that are not sufficiently cold turn into bright states and re-enter the cooling cycle. This procedure terminates when atoms in dark states have a sufficiently long lifetime, leading to a narrow peak in the momentum distribution.

The influence of stray magnetic fields on sub-Doppler cooling of alkali atoms with gray molasses on the D1 or the D2 line has never been thoroughly studied experimentally, only briefly discussed in several articles [17–21]. In these articles, similar conclusions have been reached—the lowest temperature can be achieved when the net magnetic field at the

location of atoms is near 0 G. This helps to avoid mixing of the ground-state magnetic sub-levels, which can destroy the dark state responsible for cooling. Our method is based on the observation that to a certain degree, sub-Doppler cooling in gray molasses takes place even for typical values of laboratory stray fields, which are usually below 1 G and come primarily from the Earth's magnetic field. Thus, once the sub-Doppler cooling is observed, the temperature of the cloud (or its width after ballistic expansion) can be used to minimized stray fields. For all practical purposes, the "zero field" determined this way is sufficient to enable state-of-the-art sub-Doppler cooling: without any additional magnetic field calibration, we reach $^{39}$K temperature of ~8 μK, which compares well with 6 μK reached by Salomon et al. [19] and 12 μK by Nath et al. [22].

We use three pairs of rectangular compensation coils in a near-Helmholtz configuration to generate a uniform magnetic field at the location of atoms. The characteristic parameters of each pair of coils, i.e., the inductance, the resistance and the Gauss per ampere coefficient, are determined experimentally [27] and are shown in Table 1.

**Table 1.** Characteristic parameters of each pair of rectangular coils used for generating a magnetic field vector $\vec{B}$ at the location of atoms. The last column shows the width $w$, the height $h$, and the separation $s$ between coils in each pair, given in cm. $L$—inductance, $\Omega$—resistance, and $\eta$—the conversion coefficient relating the magnetic field to the current flowing through each pair of coils.

| axis | $L$(mH) | $R(\Omega)$ | $\eta$(G/A) | $w,h,s$ |
|:---:|:---:|:---:|:---:|:---:|
| $x$ | 4.2 | 2.3 | 3.3 | 11, 26, 15 |
| $y$ | 2.9 | 2.1 | 4.4 | 15, 26, 11 |
| $z$ | 3.6 | 1.9 | 1.4 | 11, 15, 26 |

With our circuit, we are able to generate magnetic fields up to ±5.5 G along the $z$-axis. Along the remaining axes, we can easily exceed 13 G. These fields are sufficiently strong and are able to cancel most of the typical laboratory stray fields, including the Earth's magnetic field.

The magnetic field $\vec{B}$ generated by the set of three coils, each controlled by an analog voltage $U_{x,y,z}$, can be expressed as

$$\vec{B} = \begin{bmatrix} B_x \\ B_y \\ B_z \end{bmatrix} = a \begin{bmatrix} \eta_x U_x \\ \eta_y U_y \\ \eta_z U_z \end{bmatrix}, \tag{2}$$

where $a = 1.81$ A/V is the gain coefficient of the circuit and $\eta_{x,y,z}$ are the conversion coefficients between the current flowing through each pair of coils and the field the pair produces (see Table 1.)

For compensation of stray fields, we rely on measurements of the cloud width $\sigma(t_{\text{TOF}})$ after a fixed expansion time $t_{\text{TOF}}$ (typically around 8 ms). This can be treated as a proxy for the cloud's temperature due to the relation between the expansion time and the cloud's size, $\sigma^2(t_{\text{TOF}}) = \sigma_0^2 + k_{\text{B}}T/m \cdot t_{\text{TOF}}^2$, where $\sigma_0$ is the initial cloud size (here, at the end of gray molasses), $k_{\text{B}}$ is the Boltzmann constant, $m$ is the mass of the atom, and $T$ is the temperature of the cloud. For the determination of the temperature, we perform a proper time-of-flight measurement, where the cloud width is measured for various expansion times $t_{\text{TOF}}$.

Immediately before the gray molasses stage, the cloud has a temperature of ~400 μK. We note that for uncompensated stray fields present in our laboratory, the two-photon condition that provides the best cooling (here, down to 32 μK) is at the detuning of −1.6 MHz with respect to the compensated case (black diamonds in Figure 3a). With stray magnetic fields present, at the "zero magnetic field" two-photon condition, we are able to cool atoms to 70 μK. When the stray fields are compensated, we reach 8 μK.

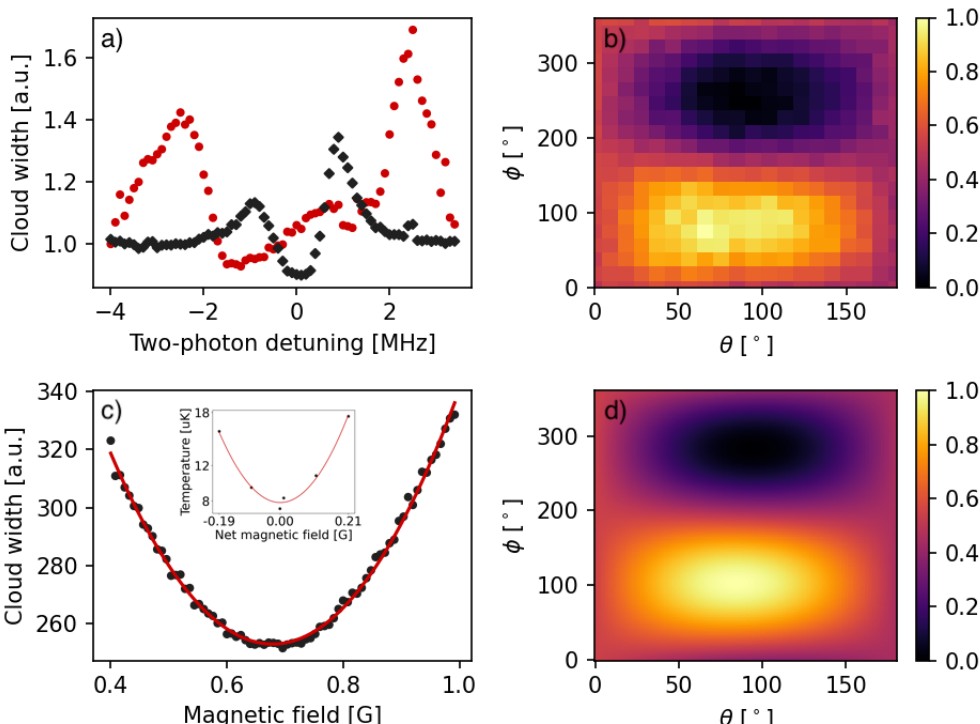

**Figure 3.** The horizontal width of the $^{39}$K cloud extracted from fitting a gaussian distribution is used as a proxy of the temperature—the narrower the cloud, the colder it is. (**a**) The width normalized to the off-resonance value after gray molasses cooling for variable Raman detuning. Red dots show the initial measurements before any compensation of stray fields. The most efficient cooling (down to 32 μK) is obtained when the repumper frequency is detuned by −1.6 MHz with respect to the "zero gauss" two-photon condition, which is used in our compensation method. Black diamonds show an analogous measurement after stray field compensation. (**b**) The width of the cloud for various tilt angles of the magnetic field vector, measured with the amplitude of the applied magnetic field fixed at $B = 0.42$ G. (**c**) The width of the cloud for various magnitudes of the *applied magnetic field* vector for $\phi_{\min} = 256°$ and $\theta_{\min} = 96°$. Inset: data verifying the quadratic dependence of temperature on the *net magnetic field* for small magnetic fields. (**d**) The predicted increase in the cloud's temperature for the same conditions as used in (**b**) as a function of tilt angles. The plot is in qualitative agreements with observations shown in (**b**). The color scale in (**b**,**c**) is normalized such that 0 and 1 correspond the minimum and maximum widths of the cloud, respectively.

We begin stray field compensation by setting the frequencies of the cooling and the repumping laser such that they fulfill the two-photon condition at 0 G, with both beams detuned by 10 MHz to the blue from $|F = 1\rangle(|F = 2\rangle) \rightarrow |F' = 2\rangle$ transitions. Next, we generate a set of analog control voltages for the bipolar sources that create a magnetic field vector $|\vec{B}| = 0.42$ G but with different tilt angles $\phi \in \langle 0, 360 \rangle$ and $\theta \in \langle 0, 180 \rangle$. The magnitude of the initially applied magnetic field is chosen based on the measurement of the stray fields with a smartphone located near the vacuum chamber. The actual value is irrelevant for our method. The angles are uniformly distributed and initially chosen such that the number of measurements is not too large. For each combination of angles $\phi$ and $\theta$, we use fluorescence imaging to obtain the width of the cloud of atoms after 8 ms ballistic expansion following a period of gray molasses cooling on the D1-line. The extent of the expanded cloud directly relates to the cloud's initial size (i.e., before sub-Doppler cooling) and can be treated as a proxy of its temperature. With 400 measurements overall, we can obtain the map shown in Figure 3b, which illustrates how the horizontal extension of the cloud changes with the tilt angles of the applied magnetic field vector. It should be emphasized that for each pair of angles shown in Figure 3b, the atoms see a different

magnetic field, which is a vector sum of the stray field and the applied external field. The data allow us to identify approximate values of angles $\phi_{\min} = 256°$ and $\theta_{\min} = 96°$ for which the cloud becomes the smallest (i.e., the coldest). For these angles, the applied field is approximately anti-aligned with the stray field (the net field is the smallest), which improves sub-Doppler cooling. Figure 3b also shows an additional feature: increased heating for $\theta_{\max} = \pi - \theta_{\min}$ and $\phi_{\max} = \phi_{\min} - \pi$, i.e., when the applied field is aligned with the stray field (the net field is the largest).

We minimize the net field by repeating the procedure devised for varying angles, but this time, the length of the vector $|\vec{B}|$ is varied, while its orientation is fixed at angles $\phi_{\min}$ and $\theta_{\min}$. The width of the cloud as a function of $|\vec{B}|$ has a minimum (Figure 3c), which we use for a second iteration of measurements, this time with angles $\phi$ and $\theta$ distributed in the range of $\pm 20°$ around the identified approximate values. Once new, more precise angles $\phi_{\min} = 256°$ and $\theta_{\min} = 92°$ are found with uncertainty of about $\pm 2°$, the length of the applied magnetic field vector is scanned again. Here, we want to probe the temperature increase near 0 G where it is not very sensitive to the magnetic field changes. To improve the quality of compensation, we extend the gray molasses stage to 1 s and monitor the total atom number vs. applied magnetic field. The further we are from the optimal cooling conditions, the more the atoms are heated, and they are eventually lost from the region of interest that we use to calculate the atom number. Under these conditions the width of the cloud also changes noticeably, but the 1 s long dynamics within gray molasses is very non-trivial and the cloud's shape becomes distorted, making the width an unreliable measure. The maximum of the atom number within a chosen region of interest allows us to identify the magnetic field that compensates stray fields the best. We want to emphasize that the long gray molasses hold time exceeds typical timescales used for efficient cooling by nearly two orders of magnitude and we believe that it is a result of careful balancing of intensities and polarizations of cooling beams. During the hold time, the cloud disperses significantly due to heating of the sample and diffusion. For compensated stray fields, exactly on two-photon resonance, the temperature increases from $\sim 8\,\mu$ to $\sim 100\,\mu$K after 1 s, and the increase is higher away from the two-photon resonance or when stray fields are not well compensated.

Several articles on sub-Doppler cooling (e.g., [18,19,28]) have reported a quadratic increase in the cloud's temperature with magnetic field during gray molasses cooling. From the inset of Figure 3c, we extract the dependence of temperature on the magnetic field, obtaining $\Delta T = 225(10)B_{\text{net}}^2\,\mu$K/G$^2$, a result comparable to $\Delta T = 300(100)B_{\text{net}}^2\,\mu$K/G$^2$ observed by Salomon et al. [19]. We plot $\Delta T$ for varying angles $\phi$ and $\theta$ (i.e., for varying net field at the location of atoms) in Figure 3d, which shows qualitative agreement with data obtained to prepare Figure 3b.

Due to the uncertainty in determining $\phi_{\min}$ and $\theta_{\min}$, for the stray field estimated at $\sim 700$ mG, we put a conservative upper limit of $\sim 100$ mG on the remaining, uncompensated field.

## 5. Conclusions

In the current study, we present a complete design and characterization of a bidirectional power supply, which is suitable for state-of-the art cold atom experiments. The power supply has a low noise at a $10^{-5}$ ppm level and can be used to drive most of the typical coils used for compensating stray magnetic fields. Due to the presence of a smooth zero current crossing, it enables adiabatic change in the quantization axis in any desired direction as well as generation of magnetic fields with a DC offset oscillating at frequencies up to $\sim 2$ kHz. With optimized lead and lag compensators, the circuit can be used to create a time-orbiting trap [29], an important tool for generation of Bose–Einstein condensates of magnetically trapped atoms [3] or in experiments with dipolar quantum gases where rotating magnetic fields are of interest [30]. The ability to change the direction of the magnetic field vector would enable the control of the anisotropy of interactions in dipolar

species possessing low-field Feshbach resonances, such as erbium [31], dysprosium [32,33], and mixtures of erbium and dysprosium [34].

We use the designed power supply to demonstrate a fast and robust method of compensating stray magnetic fields by minimizing the temperature of a cloud of $^{39}$K atoms after a period of sub-Doppler cooling in gray molasses. We exploit the fact that sub-Doppler cooling in gray molasses takes place already for typical laboratory magnetic fields, and it can be optimized as a function of the direction and amplitude of the magnetic field vector. The usefulness of the presented method becomes especially apparent during initial stages of building a new experimental setup, when it is often more important to optimize various cooling stages than to precisely know the value of the field. This method does not require microwave or radio frequency sources to drive Zeeman or hyperfine transitions for magnetic field calibration. Even if such sources are implemented in a setup, a reasonable guess of what the field value is makes it useful for more precise stray field cancellation methods based on driving rf/microwave or Raman transitions [35], which is usually much more time consuming. Since D1-line gray molasses cooling has been already demonstrated for all commonly laser-cooled alkali atoms, the present method could be appealing to many research groups.

**Supplementary Materials:** The following are available online at https://www.mdpi.com/article/10.3390/app112110474/s1. Schematics of the bidirectional current source, Gerber files for manufacturing purposes and the bill of materials.

**Author Contributions:** J.D. designed, constructed, and characterized the bipolar power supply; M.B. performed optimization of sub-Doppler cooling; M.S. initiated and supervised the project, devised the measurement plan, acquired funding, and analyzed and interpreted data; M.S. and J.D. wrote the manuscript with input from M.B. All authors have read and agreed to the published version of the manuscript.

**Funding:** This research was funded by the Foundation for Polish Science within the Homing program and the National Science Centre of Poland (grant No. 2016/21/D/ST2/02003 and a postdoctoral fellowship for M.S., grant No. DEC-2015/16/S/ST2/00425).

**Institutional Review Board Statement:** Not applicable.

**Informed Consent Statement:** Not applicable.

**Data Availability Statement:** The data presented in this study and related design files for the bidirectional current source are available on request from the corresponding author (J.D.).

**Acknowledgments:** We would like to acknowledge Paweł Arciszewski for his assistance during the early stage of setting up the experiment and Paweł Kowalczyk for his support.

**Conflicts of Interest:** The authors declare no conflict of interest.

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
