# Peer review of "Bidirectional, Analog Current Source Benchmarked with Gray Molasses-Assisted Stray Magnetic Field Compensation"

_applsci, doi:10.3390/app112110474_

Round 1

Reviewer 1 Report

This article describes developments of a bi-directional analog currents source for pair of coils, which are suitable for state-of-the-art Doppler cooling experiments. It is well described and sounds good. However, the following points should be considered and explained in the article. 

1) In Fig.1: Units of components should be given. For example, kΩ, mH, pF.

    What are 1m5 and 1R?

2) Line 191 and Fig.2: 1 Ap-p should be Ip-p=1 A. 

3) Line 194: "both methods agreeing, very well" Please evaluate quantitatively.

4) In figure caption of Fig.2: "The zero is smooth" Please explain where the crossing is.

5) Line 247: Please give the used equation of cloud and temperature. In Fig. 3 a) and c), the cloud width is given in arbitrary units. How could you calculate the absolute temperature of atoms?

6) Line 305: "225(10) " How do you estimate the standard deviation?

Reviewer 2 Report

The authors have presented an experimental system that operates as a low-noise current source. The utility of the source is demonstrated for power electromagnetic coils that are used to null the ambient fields a laser-cooled sample of potassium atoms for enhanced gray molasses.

In general the paper is well written and clear in stating the objectives, the methods, and the results. It will find an obvious readership amongst the laser-cooling community but will have value in the wider experimental community. For this reason I recommend it be accepted. I offer some minor comments that I hope would add value to the paper

Line 46. There are a large number of low-cost ADCs. At card level, the PCIE-1802L-AE offers 4 channels of 24-bit control for <€2k with >200kS/channel. At a component level, Analog, for example, offer a number of devices with ≥24-bit resolution. In an evaluation board these are quite affordable, e.g. EVAL-AD4115SDZ is ≤€100

Line 47. It might be worth referencing digitally-controlled current drivers that have shown good performance. For example, from the Durfee group - http://dx.doi.org/10.1063/1.3630950

Line 93. Using Eqn 1 and with parameters in Fig 1 seems to give a transfer 2A/V

Line 144. The single-sided supply is a significant advantage and could be emphasised earlier

Line 161. I am confused by the statement that the FFT is proportional to the PSD. Is it not that the PSD ~ |FFT|^2 ?

Line 169. Is the word “measured” missing? “… noise level has been measured to be of …”

Line 174. The thermal management is mentioned latter. However, would the authors expect the system to work at the higher currents of 8A with presumably four times the heat dissipation?

Figure 2. I was surprised that the oscillating current appears to span the range 0-1A. The caption references smooth zero crossings, but these are not clearly evident. If they have data for a signal that is symmetric about 0A then that would be more useful.

It is difficult to draw any comparison between the data in Fig.2(b) and (c). In place of (c) I would suggest that plots of the difference of the sensors, the ratio of the sensors, or even a parametric plot of monitor vs transducer would be useful.

Line 197. What are the simulated values of the roll-off frequencies? It would be good to see some short discussion here, particularly comparing the values for the purely passive L-R parameters of the coils. The values from Table 1 give roll off frequencies of {87, 115, 84} Hz for the different coils.

Line 248. Make a statement that rectangular coils are being used.

Figure 3.

The figure appears far in advance of where it is discussed in the text. This caused some confusion about details that would be removed if the figure was move a page or so later.

Parts (b) and (c) need colorbars.

Change “lengths” to “magnitudes” in discussion of part (c) in the caption

Line 263. How are the temperatures measured here (400 µK) and later?

Line 277. The cloud size after time of flight can be used as a proxy for temperature. However, the expanded cloud size needs to be large relative to the initial cloud size; i.e, such that the thermal expansion is sufficient to dominate over the initial distribution. It is important also to note that gray molasses is known to increase number density, and hence reduces the initial cloud size.

Line 298. Is the 1-second duration a type? Most gray molasses experiments report durations of approximately 10ms. 

Line 319. TOP traps are typically driven at 10kHz, or higher. Can this be implemented just by tuning the lead and lag compensation?

General comments: 

  • I suggest adding reference to gray molasses in the title; approximately half the paper is dedicated to this aspect of the work.
  • The application of the circuit for nulling fields is clearly useful. However, the data show that stability at the level of 1e-5 is not needed for this application. Could the authors add comment on experimental situations require this level of control. One example might be for Feshbach resonances, which are of interest in 39K, albeit the lowest I’m aware of is at 26G, although I do not know this atom well – if the authors are aware of other, lower, resonances, then this could be included. The system, running at full capacity on all channels seems to reach a net field of ~22G. However, if the current was increased to 6A then the y-axis alone could achieve 26G
  • Many, if not most, experimental groups that I know of null magnetic fields using iterative scans of the field along three orthogonal axes. Some groups also null gradient fields in this method. Microwave spectroscopy is indeed slow and expensive and, to my knowledge, is rarely used except for BEC or lattice experiments. The authors perhaps could make clear that using atomic temperature to null field is a standard technique. However, the method of the authors, using the angle scans in Fig.3, is not one I had come across before. It also perhaps provides an intuitive picture for readers.
